# Approximation and Convergence Properties of Generative Adversarial Learning

**Shuang Liu**
University of California, San Diego
shuangliu@ucsd.edu

**Olivier Bousquet**
Google Brain
obousquet@google.com

**Kamalika Chaudhuri**
University of California, San Diego
kamalika@cs.ucsd.edu

## Abstract

Generative adversarial networks (GAN) approximate a target data distribution by jointly optimizing an objective function through a "two-player game" between a generator and a discriminator. Despite their empirical success, however, two very basic questions on how well they can approximate the target distribution remain unanswered. First, it is not known how restricting the discriminator family affects the approximation quality. Second, while a number of different objective functions have been proposed, we do not understand when convergence to the global minima of the objective function leads to convergence to the target distribution under various notions of distributional convergence.

In this paper, we address these questions in a broad and unified setting by defining a notion of adversarial divergences that includes a number of recently proposed objective functions. We show that if the objective function is an adversarial divergence with some additional conditions, then using a restricted discriminator family has a moment-matching effect. Additionally, we show that for objective functions that are strict adversarial divergences, convergence in the objective function implies weak convergence, thus generalizing previous results.

## 1 Introduction

Generative adversarial networks (GANs) have attracted an enormous amount of recent attention in machine learning. In a generative adversarial network, the goal is to produce an approximation to a target data distribution $\mu$ from which only samples are available. This is done iteratively via two components – a generator and a discriminator, which are usually implemented by neural networks. The generator takes in random (usually Gaussian or uniform) noise as input and attempts to transform it to match the target distribution $\mu$; the discriminator aims to accurately discriminate between samples from the target distribution and those produced by the generator. Estimation proceeds by iteratively refining the generator and the discriminator to optimize an objective function until the target distribution is indistinguishable from the distribution induced by the generator. The practical success of GANs has led to a large volume of recent literature on variants which have many desirable properties; examples are the f-GAN [10], the MMD-GAN [5, 9], the Wasserstein-GAN [2], among many others.

In spite of their enormous practical success, unlike more traditional methods such as maximum likelihood inference, GANs are theoretically rather poorly-understood. In particular, two very basic questions on how well they can *approximate* the target distribution $\mu$, even in the presence of a very large number of samples and perfect optimization, remain largely unanswered. The first relates to the

role of the discriminator in the quality of the approximation. In practice, the discriminator is usually restricted to belong to some family, and it is not understood in what sense this restriction affects the distribution output by the generator. The second question relates to convergence; different variants of GANs have been proposed that involve different objective functions (to be optimized by the generator and the discriminator). However, it is not understood under what conditions minimizing the objective function leads to a good approximation of the target distribution. More precisely, does a sequence of distributions output by the generator that converges to the global minimum under the objective function always converge to the target distribution $\mu$ under some standard notion of distributional convergence?

In this work, we consider these two questions in a broad setting. We first characterize a very general class of objective functions that we call *adversarial divergences*, and we show that they capture the objective functions used by a variety of existing procedures that include the original GAN [7], f-GAN [10], MMD-GAN [5, 9], WGAN [2], improved WGAN [8], as well as a class of entropic regularized optimal transport problems [6]. We then define the class of *strict adversarial divergences* – a subclass of adversarial divergences where the minimizer of the objective function is uniquely the target distribution. This characterization allows us to address the two questions above in a unified setting, and translate the results to an entire class of GANs with little effort.

First, we address the role of the discriminator in the approximation in Section 4. We show that if the objective function is an adversarial divergence that obeys certain conditions, then using a restricted class of discriminators has the effect of matching *generalized moments*. A concrete consequence of this result is that in linear f-GANs, where the discriminator family is the set of all affine functions over a vector $\psi$ of features maps, and the objective function is an f-GAN, the optimal distribution $\nu$ output by the GAN will satisfy $\mathbb{E}_{x\sim\mu}[\psi(x)] = \mathbb{E}_{x\sim\nu}[\psi(x)]$ regardless of the specific $f$-divergence chosen in the objective function. Furthermore, we show that a neural network GAN is just a supremum of linear GANs, therefore has the same moment-matching effect.

We next address convergence in Section 5. We show that convergence in an adversarial divergence implies some standard notion of topological convergence. Particularly, we show that provided an objective function is a strict adversarial divergence, convergence to $\mu$ in the objective function implies weak convergence of the output distribution to $\mu$. While convergence properties of some isolated objective functions were known before [2], this result extends them to a broad class of GANs. An additional consequence of this result is the observation that as the Wasserstein distance metrizes weak convergence of probability distributions (see e.g. [14]), Wasserstein-GANs have the weakest[1] objective functions in the class of strict adversarial divergences.

## 2 Notations

We use bold constants (e.g., $\mathbf{0}, \mathbf{1}, \mathbf{x_0}$) to denote constant functions. We denote by $f \circ g$ the function composition of $f$ and $g$. We denote by $Y^X$ the set of functions maps from the set $X$ to the set $Y$. We denote by $\mu \otimes \nu$ the product measure of $\mu$ and $\nu$. We denote by $\text{int}(X)$ the interior of the set $X$. We denote by $\mathbb{E}_\mu[f]$ the integral of $f$ with respect to measure $\mu$.

Let $f : \mathbb{R} \to \mathbb{R} \cup \{+\infty\}$ be a convex function, we denote by $\text{dom} f$ the effective domain of $f$, that is, $\text{dom} f = \{x \in \mathbb{R}, f(x) < +\infty\}$; and we denote by $f^*$ the convex conjugate of $f$, that is, $f^*(x^*) = \sup_{x\in\mathbb{R}} \{x^* \cdot x - f(x)\}$.

For a topological space $\Omega$, we denote by $C(\Omega)$ the set of continuous functions on $\Omega$, $C_b(\Omega)$ the set of bounded continuous functions on $\Omega$, $rca(\Omega)$ the set of finite signed regular Borel measures on $\Omega$, and $\mathcal{P}(\Omega)$ the set of probability measures on $\Omega$.

Given a non-empty subspace $Y$ of a topological space $X$, denote by $X/Y$ the quotient space equipped with the quotient topology $\sim_Y$, where for any $a, b \in X$, $a \sim_Y b$ if and only if $a = b$ or $a, b$ both belong to $Y$. The equivalence class of each element $a \in X$ is denoted as $[a] = \{b : a \sim_Y b\}$.

# 3 General Framework

Let $\mu$ be the target data distribution from which we can draw samples. Our goal is to find a generative model $\nu$ to approximate $\mu$. Informally, most GAN-style algorithms model this approximation as solving the following problem

$$\inf_{\nu} \sup_{f \in \mathcal{F}} \mathbb{E}_{x \sim \mu, \, y \sim \nu} \left[ f(x, y) \right],$$

where $\mathcal{F}$ is a class of functions. The process is usually considered *adversarial* in the sense that it can be thought of as a two-player minimax game, where a *generator* $\nu$ is trying to mimick the true distribution $\mu$, and a *adversary* $f$ is trying to distinguish between the true and generated distributions. However, another way to look at it is as the minimization of the following objective function

$$\nu \;\longmapsto\; \sup_{f \in \mathcal{F}} \mathbb{E}_{x \sim \mu, \, y \sim \nu} \left[ f(x, y) \right] \tag{1}$$

This objective function measures how far the target distribution $\mu$ is from the current estimate $\nu$. Hence, minimizing this function can lead to a good approximation of the target distribution $\mu$.

This leads us to the concept of *adversarial divergence*.

**Definition 1** (Adversarial divergence). *Let $X$ be a topological space, $\mathcal{F} \subseteq C_b(X^2)$, $\mathcal{F} \neq \emptyset$. An adversarial divergence $\tau$ over $X$ is a function*

$$
\begin{aligned}
\mathcal{P}(X) \times \mathcal{P}(X) &\longrightarrow \mathbb{R} \cup \{+\infty\} \\
(\mu, \nu) &\longmapsto \tau(\mu \| \nu) = \sup_{f \in \mathcal{F}} \mathbb{E}_{\mu \otimes \nu} \left[ f \right].
\end{aligned}
\tag{2}
$$

Observe that in Definition 1 if we have a fixed target distribution $\mu$, then (2) is reduced to the objective function (1). Also, notice that because $\tau$ is the supremum of a family of linear functions (in each of the variables $\mu$ and $\nu$ separately), it is convex in each of its variables.

Definition 1 captures the objective functions used by a variety of existing GAN-style procedures. In practice, although the function class $\mathcal{F}$ can be complicated, it is usually a transformation of a simple function class $\mathcal{V}$, which is the set of *discriminators* or *critics*, as they have been called in the GAN literature. We give some examples by specifying $\mathcal{F}$ and $\mathcal{V}$ for each objective function.

(a) GAN [7].
$$\mathcal{F} = \{x, y \mapsto \log(u(x)) + \log(1 - u(y)) : u \in \mathcal{V}\}$$
$$\mathcal{V} = (0, 1)^X \cap C_b(X).$$

(b) $f$-GAN [10]. Let $f : \mathbb{R} \to \mathbb{R} \cup \{\infty\}$ be a convex lower semi-continuous function. Assume $f^*(x) \geq x$ for any $x \in \mathbb{R}$, $f^*$ is continuously differentiable on $\operatorname{int}(\operatorname{dom} f^*)$, and there exists $x_0 \in \operatorname{int}(\operatorname{dom} f^*)$ such that $f^*(x_0) = x_0$.
$$\mathcal{F} = \{x, y \mapsto v(x) - f^*(v(y)) : v \in \mathcal{V}\},$$
$$\mathcal{V} = (\operatorname{dom} f^*)^X \cap C_b(X).$$

(c) MMD-GAN [5, 9]. Let $k : X^2 \to \mathbb{R}$ be a universal reproducing kernel. Let $\mathcal{M}$ be the set of signed measures on $X$.
$$\mathcal{F} = \{x, y \mapsto v(x) - v(y) : v \in \mathcal{V}\},$$
$$\mathcal{V} = \left\{x \mapsto \mathbb{E}_{\mu} \left[ k(x, \cdot) \right] : \mu \in \mathcal{M}, \, \mathbb{E}_{\mu^2}[k] \leq 1 \right\}.$$

(d) Wasserstein-GAN (WGAN) [2]. Assume $X$ is a metric space.
$$\mathcal{F} = \{x, y \mapsto v(x) - v(y) : v \in \mathcal{V}\},$$
$$\mathcal{V} = \left\{v \in C_b(X) : \|v\|_{\mathrm{Lip}} \leq K \right\},$$
where $K$ is a positive constant, $\|\cdot\|_{\mathrm{Lip}}$ denotes the Lipschitz constant.

(e) WGAN-GP (Improved WGAN) [8]. Assume $X$ is a convex subset of a Euclidean space.
$$\mathcal{F} = \{x, y \mapsto v(x) - v(y) - \eta \mathbb{E}_{t \sim U} \left[ (\|\nabla v(tx + (1 - t)y)\|_2 - 1)^p \right] : v \in \mathcal{V}\},$$
$$\mathcal{V} = C^1(X),$$
where $U$ is the uniform distribution on $[0, 1]$, $\eta$ is a positive constant, $p \in (1, \infty)$.

(f) (Regularized) Optimal Transport [6]. [2] Let $c : X^2 \to \mathbb{R}$ be some transportation cost function, $\epsilon \geq 0$ be the strength of regularization. If $\epsilon = 0$ (no regularization), then

$$\mathcal{F} = \left\{ x, y \mapsto u(x) + v(y) : (u, v) \in \mathcal{V} \right\}, \tag{3}$$
$$\mathcal{V} = \left\{ (u, v) \in C_b(X) \times C_b(X), \ u(x) + v(y) \leq c(x, y) \text{ for any } x, y \in X \right\};$$

if $\epsilon > 0$, then

$$\mathcal{F} = \left\{ x, y \mapsto u(x) + v(y) - \epsilon \exp\left( \frac{u(x) + v(y) - c(x, y)}{\epsilon} \right) : u, v \in \mathcal{V} \right\}, \tag{4}$$
$$\mathcal{V} = C_b(X).$$

In order to study an adversarial divergence $\tau$, it is critical to first understand at which points the divergence is minimized. More precisely, let $\tau$ be an adversarial divergence and $\mu^*$ be the target probability measure. We are interested in the set of probability measures that minimize the divergence $\tau$ when the first argument of $\tau$ is set to $\mu^*$, i.e., the set $\arg\min \tau(\mu^* \| \cdot) = \{ \mu : \tau(\mu^* \| \mu) = \inf_\nu \tau(\mu^* \| \nu) \}$. Formally, we define the set $\mathrm{OPT}_{\tau, \mu^*}$ as follows.

**Definition 2** ($\mathrm{OPT}_{\tau, \mu^*}$). *Let $\tau$ be an adversarial divergence over a topological space $X$, $\mu^* \in \mathcal{P}(X)$. Define $\mathrm{OPT}_{\tau, \mu^*}$ to be the set of probability measures that minimize the function $\tau(\mu^* \| \cdot)$. That is,*

$$\mathrm{OPT}_{\tau, \mu^*} \triangleq \left\{ \mu \in \mathcal{P}(X) : \tau(\mu^* \| \mu) = \inf_{\mu' \in \mathcal{P}(X)} \tau(\mu^* \| \mu') \right\}.$$

Ideally, the target probability measure $\mu^*$ should be one and the only one that minimizes the objective function. The notion of *strict adversarial divergence* captures this property.

**Definition 3** (Strict adversarial divergence). *Let $\tau$ be an adversarial divergence over a topological space $X$, $\tau$ is called a strict adversarial divergence if for any $\mu^* \in \mathcal{P}(X)$, $\mathrm{OPT}_{\tau, \mu^*} = \{\mu^*\}$.*

For example, if the underlying space $X$ is a compact metric space, then examples (c) and (d) induce metrics on $\mathcal{P}(X)$ (see, e.g., [12]), therefore are strict adversarial divergences.

In the next two sections, we will answer two questions regarding the set $\mathrm{OPT}_{\tau, \mu^*}$: how well do the elements in $\mathrm{OPT}_{\tau, \mu^*}$ approximate the target distribution $\mu^*$ when restricting the class of discriminators? (Section 4); and does a sequence of distributions that converges in an adversarial divergence also converges to $\mathrm{OPT}_{\tau, \mu^*}$ under some standard notion of distributional convergence? (Section 5)

## 4 Generalized Moment Matching

To motivate the discussion in this section, recall example (b) in Section 3. It can be shown that under some mild conditions, $\tau$, the objective function of $f$-GAN, is actually the $f$-divergence, and the minimizer of $\tau(\mu^* \| \cdot)$ is only $\mu^*$ [10]. However, in practice, the discriminator class $\mathcal{V}$ is usually implemented by a feedforward neural network, and it is known that a fixed neural network has limited capacity (e.g., it cannot implement the set of all the bounded continuous function). Therefore, one could ask what will happen if we restrict $\mathcal{V}$ to a sub-class $\mathcal{V}'$? Obviously one would expect $\mu^*$ not be the unique minimizer of $\tau(\mu^* \| \cdot)$ anymore, that is, $\mathrm{OPT}_{\tau, \mu^*}$ contains elements other than $\mu^*$. What can we say about the elements in $\mathrm{OPT}_{\tau, \mu^*}$ now? Are all of them close to $\mu^*$ in a certain sense? In this section we will answer these questions.

More formally, we consider $\mathcal{F} = \{ m_\theta - r_\theta : \theta \in \Theta \}$ to be a function class indexed by a set $\Theta$. We can think of $\Theta$ as the parameter set of a feedforward neural network. Each $m_\theta$ is thought to be a matching between two distributions, in the sense that $\mu$ and $\nu$ are matched under $m_\theta$ if and only if $\mathbb{E}_{\mu \otimes \nu}[m_\theta] = 0$. In particular, if each $m_\theta$ is corresponding to some function $v_\theta$ such that $m_\theta(x, y) = v_\theta(x) - v_\theta(y)$, then $\mu$ and $\nu$ are matched under $m_\theta$ if and only if some generalized moment of $\mu$ and $\nu$ are equal: $\mathbb{E}_\mu[v_\theta] = \mathbb{E}_\nu[v_\theta]$. Each $r_\theta$ can be thought as a residual.

We will now relate the matching condition to the optimality of the divergence. In particular, define

$$\mathcal{M}_{\mu^*} \triangleq \{ \mu : \forall \theta \in \Theta, \ \mathbb{E}_{\mu^*}[v_\theta] = \mathbb{E}_\mu[v_\theta] \},$$

We will give sufficients conditions for members of $\mathcal{M}_{\mu^*}$ to be in $\mathrm{OPT}_{\tau, \mu^*}$.

**Theorem 4.** *Let $X$ be a topological space, $\Theta \subseteq \mathbb{R}^n$, $\mathcal{V} = \{v_\theta \in C_b(X) : \theta \in \Theta\}$, $\mathcal{R} = \{r_\theta \in C_b(X^2) : \theta \in \Theta\}$. Let $m_\theta(x,y) = v_\theta(x) - v_\theta(y)$. If there exists $c \in \mathbb{R}$ such that for any $\mu, \nu \in \mathcal{P}(X)$, $\inf_{\theta \in \Theta} \mathbb{E}_{\mu \otimes \nu}[r_\theta] = c$ and there exists some $\theta_\nu^\mu \in \Theta$ such that $\mathbb{E}_{\mu \otimes \nu}[r_{\theta_\nu^\mu}] = c$ and $\mathbb{E}_{\mu \otimes \nu}[m_{\theta_\nu^\mu}] \geq 0$, then $\tau(\mu \| \nu) = \sup_{\theta \in \Theta} \mathbb{E}_{\mu \otimes \nu}[m_\theta - r_\theta]$ is an adversarial divergence over $X$ and for any $\mu^* \in \mathcal{P}(X)$, $\mathrm{OPT}_{\tau, \mu^*} \supset \mathcal{M}_{\mu^*}$.*

We now review the examples (a)-(e) in Section 3, show how to write each $f \in \mathcal{F}$ into $m_\theta - r_\theta$, and specify $\theta_\nu^\mu$ in each case such that the conditions of Theorem 4 can be satisfied.

(a) GAN. Note that for any $x \in (0,1)$, $\log(1/(x(1-x))) \geq \log(4)$. Let $u_{\theta_\nu^\mu} = \frac{1}{2}$,

$$
\begin{aligned}
f_\theta(x,y) &= \log(u_\theta(x)) + \log(1 - u_\theta(y)) \\
&= \underbrace{\log(u_\theta(x)) - \log(u_\theta(y))}_{m_\theta(x,y)\,\left(\text{note }\mathbb{E}_{\mu \otimes \nu}\left[m_{\theta_\nu^\mu}\right] = 0\right)} - \underbrace{\log\left(1/\left(u_\theta(y)(1 - u_\theta(y))\right)\right)}_{r_\theta(x,y)\,\left(\text{note }r_\theta(x,y) \geq r_{\theta_\nu^\mu}(x,y) = \log(4)\right)} \quad .
\end{aligned}
$$

(b) $f$-GAN. Recall that $f^*(x) - x \geq 0$ for any $x \in \mathbb{R}$ and $f^*(x_0) = x_0$. Let $v_{\theta_\nu^\mu} = \mathbf{x_0}$,

$$
\begin{aligned}
f_\theta(x,y) &= v_\theta(x) - f^*(v_\theta(y)) \\
&= \underbrace{v_\theta(x) - v_\theta(y)}_{m_\theta(x,y)\,\left(\text{note }\mathbb{E}_{\mu \otimes \nu}\left[m_{\theta_\nu^\mu}\right] = 0\right)} - \underbrace{(f^*(v_\theta(y)) - v_\theta(y))}_{r_\theta(x,y)\,\left(\text{note }r_\theta(x,y) \geq r_{\theta_\nu^\mu}(x,y) = 0\right)} \quad .
\end{aligned} \tag{5}
$$

(c, d) MMD-GAN or Wasserstein-GAN. Let $v_{\theta_\nu^\mu} = \mathbf{0}$,

$$
f_\theta(x,y) = \underbrace{v_\theta(x) - v_\theta(y)}_{m_\theta(x,y)\,\left(\text{note }\mathbb{E}_{\mu \otimes \nu}\left[m_{\theta_\nu^\mu}\right] = 0\right)} - \underbrace{0}_{r_\theta(x,y)\,\left(\text{note }r_\theta(x,y) = r_{\theta_\nu^\mu}(x,y) = 0\right)} \quad .
$$

(e) WGAN-GP. Note that the function $x \mapsto x^p$ is nonnegative on $\mathbb{R}$. Let

$$
v_{\theta_\nu^\mu} = \begin{cases} (x_1, x_2, \cdots, x_n) \mapsto \frac{\sum_{i=1}^n x_i}{\sqrt{n}}, & \text{if } \mathbb{E}_\mu[\sum_{i=1}^n x_i] \geq \mathbb{E}_\nu[\sum_{i=1}^n x_i], \\ (x_1, x_2, \cdots, x_n) \mapsto -\frac{\sum_{i=1}^n x_i}{\sqrt{n}}, & \text{otherwise,} \end{cases}
$$

$$
f_\theta(x,y) = \underbrace{v_\theta(x) - v_\theta(y)}_{m_\theta(x,y)\,\left(\text{note }\mathbb{E}_{\mu \otimes \nu}\left[m_{\theta_\nu^\mu}\right] \geq 0\right)} - \underbrace{\eta \mathbb{E}_{t \sim U}\left[(\|\nabla v(tx + (1-t)y)\|_2 - 1)^p\right]}_{r_\theta(x,y)\,\left(\text{note }r_\theta(x,y) \geq r_{\theta_\nu^\mu}(x,y) = 0\right)} \,.
$$

We now refine the previous result and show that under some additional conditions on $m_\theta$ and $r_\theta$, the optimal elements of $\tau$ are fully characterized by the matching condition, i.e. $\mathrm{OPT}_{\tau, \mu^*} = \mathcal{M}_{\mu^*}$.

**Theorem 5.** *Under the assumptions of Theorem 4, if $\theta_\nu^\mu \in \mathrm{int}(\Theta)$ and both $\theta \mapsto \mathbb{E}_{\mu \otimes \nu}[m_\theta]$ and $\theta \mapsto \mathbb{E}_{\mu \otimes \nu}[r_\theta]$ have gradients at $\theta_\nu^\mu$, and*

$$
\left(\mathbb{E}_{\mu \otimes \nu}[m_{\theta_\nu^\mu}] = 0 \text{ and } \exists \theta', \ \mathbb{E}_{\mu \otimes \nu}[m_{\theta'}] \neq 0\right) \implies \nabla_{\theta_\nu^\mu} \mathbb{E}_{\mu \otimes \nu}[m] \neq \mathbf{0}. \tag{6}
$$

*Then for any $\mu^* \in \mathcal{P}(X)$, $\mathrm{OPT}_{\tau, \mu^*} = \mathcal{M}_{\mu^*}$.*

We remark that Theorem 4 is relatively intuitive, while Theorem 5 requires extra conditions, and is quite counter-intuitive especially for algorithms like $f$-GANs.

## 4.1 Example: Linear $f$-GAN

We first consider a simple algorithm called *linear $f$-GAN*. Suppose we are provided with a feature map $\psi$ that maps each point $x$ in the sample space $X$ to a feature vector $(\psi_1(x), \psi_2(x), \cdots, \psi_n(x))$ where each $\psi_i \in C_b(X)$. We are satisfied that any distribution $\mu$ is a good approximation of the target distribution $\mu^*$ as long as $\mathbb{E}_{\mu^*}[\psi] = \mathbb{E}_\mu[\psi]$. For example, if $X \subseteq \mathbb{R}$ and $\psi_k(x) = x^k$, to say $\mathbb{E}_{\mu^*}[\psi] = \mathbb{E}_\mu[\psi]$ is equivalent to say the first $n$ moments of $\mu^*$ and $\mu$ are matched. Recall that in the standard $f$-GAN (example (b) in Section 3), $\mathcal{V} = (\mathrm{dom}\, f^*)^X \cap C_b(X)$. Now instead of using the discriminator class $\mathcal{V}$, we use a restricted discriminator class $\mathcal{V}' \subseteq \mathcal{V}$, containing the linear (or more precisely, affine) transformations of $\psi$ – the set $\mathcal{V}' = \{\theta^\mathsf{T}(\psi, \mathbf{1}) : \theta \in \Theta\} \subseteq \mathcal{V}$, where $\Theta = \{\theta \in \mathbb{R}^{n+1} : \forall x \in X, \ \theta^\mathsf{T}(\psi(x), 1) \in \mathrm{dom}\, f^*\}$. We will show that now $\mathrm{OPT}_{\tau, \mu^*}$ contains exactly those $\mu$ such that $\mathbb{E}_{\mu^*}[\psi] = \mathbb{E}_\mu[\psi]$, regardless of the specific $f$ chosen. Formally,

**Corollary 6** (linear $f$-GAN)**.** *Let $X$ be a compact topological space. Let $f$ be a function as defined in example (b) of Section 3. Let $\psi = (\psi_i)_{i=1}^n$ be a vector of continuously differentiable functions on $X$. Let $\Theta = \left\{ \theta \in \mathbb{R}^{n+1} : \forall x \in X, \; \theta^{\mathsf{T}}(\psi(x), 1) \in \mathrm{dom}\, f^* \right\}$. Let $\tau$ be the objective function of the linear $f$-GAN*

$$\tau(\mu||\nu) = \sup_{\theta \in \Theta} \left( \mathbb{E}_\mu[\theta^{\mathsf{T}}(\psi, \mathbf{1})] - \mathbb{E}_\nu[f^* \circ (\theta^{\mathsf{T}}(\psi, \mathbf{1}))] \right).$$

*Then for any $\mu^* \in \mathcal{P}(X)$, $\mathrm{OPT}_{\tau, \mu^*} = \{\mu : \tau(\mu^*||\mu) = 0\} = \{\mu : \mathbb{E}_{\mu^*}[\psi] = \mathbb{E}_\mu[\psi]\} \ni \mu^*$.*

A very concrete example of Corollary 6 could be, for example, the linear KL-GAN, where $f(u) = u \log u$, $f^*(t) = \exp(t-1)$, $\psi = (\psi_i)_{i=1}^n$, $\Theta = \mathbb{R}^{n+1}$. The objective function is

$$\tau(\mu||\nu) = \sup_{\theta \in \mathbb{R}^{n+1}} \left( \mathbb{E}_\mu[\theta^{\mathsf{T}}(\psi, \mathbf{1})] - \mathbb{E}_\nu[\exp(\theta^{\mathsf{T}}(\psi, \mathbf{1}) - 1)] \right),$$

### 4.2 Example: Neural Network $f$-GAN

Next we consider a more general and practical example: an $f$-GAN where the discriminator class $\mathcal{V}' = \{v_\theta : \theta \in \Theta\}$ is implemented through a feedforward neural network with weight parameter set $\Theta$. We assume that all the activation functions are continuously differentiable (e.g., sigmoid, tanh), and the last layer of the network is a linear transformation plus a bias. We also assume $\mathrm{dom}\, f^* = \mathbb{R}$ (e.g., the KL-GAN where $f^*(t) = \exp(t-1)$).

Now observe that when all the weights before the last layer are fixed, the last layer acts as a discriminator in a *linear $f$-GAN*. More precisely, let $\Theta_{pre}$ be the index set for the weights before the last layer. Then each $\theta_{\mathrm{pre}} \in \Theta_{\mathrm{pre}}$ corresponds to a feature map $\psi^{\theta_{\mathrm{pre}}}$. Let the linear $f$-GAN that corresponds to $\psi^{\theta_{\mathrm{pre}}}$ be $\tau_{\theta_{\mathrm{pre}}}$, the adversarial divergence induced by the Neural Network $f$-GAN is

$$\tau(\mu^*||\mu) = \sup_{\theta_{\mathrm{pre}} \in \Theta_{\mathrm{pre}}} \tau_{\theta_{\mathrm{pre}}}(\mu^*||\mu)$$

Clearly $\mathrm{OPT}_{\tau, \mu^*} \supseteq \bigcap_{\theta_{\mathrm{pre}} \in \Theta_{\mathrm{pre}}} \mathrm{OPT}_{\tau_{\theta_{\mathrm{pre}}}, \mu^*}$. For the other direction, note that by Corollary 6, for any $\theta_{\mathrm{pre}} \in \Theta_{\mathrm{pre}}$, $\tau_{\theta_{\mathrm{pre}}}(\mu^*||\mu) \geq 0$ and $\tau_{\theta_{\mathrm{pre}}}(\mu^*||\mu^*) = 0$. Therefore $\tau(\mu^*||\mu) \geq 0$ and $\tau(\mu^*||\mu^*) = 0$. If $\mu \in \mathrm{OPT}_{\tau, \mu^*}$, then $\tau(\mu^*||\mu) = 0$. As a consequence, $\tau_{\theta_{\mathrm{pre}}}(\mu^*||\mu) = 0$ for any $\theta_{\mathrm{pre}} \in \Theta_{\mathrm{pre}}$. Therefore $\mathrm{OPT}_{\tau, \mu^*} \subseteq \bigcap_{\theta_{\mathrm{pre}} \in \Theta_{\mathrm{pre}}} \mathrm{OPT}_{\tau_{\theta_{\mathrm{pre}}}, \mu^*}$. Therefore, by Corollary 6,

$$\mathrm{OPT}_{\tau, \mu^*} = \bigcap_{\theta_{\mathrm{pre}} \in \Theta_{\mathrm{pre}}} \mathrm{OPT}_{\tau_{\theta_{\mathrm{pre}}}, \mu^*} = \{\mu : \forall \theta \in \Theta, \; \mathbb{E}_{\mu^*}[v_\theta] = \mathbb{E}_\mu[v_\theta]\}.$$

That is, the minimizer of the Neural Network $f$-GAN are exactly those distributions that are indistinguishable under the expectation of any discriminator network $v_\theta$.

## 5 Convergence

To motivate the discussion in this section, consider the following question. Let $\delta_{x_0}$ be the delta distribution at $x_0 \in \mathbb{R}$, that is, $x = x_0$ with probability 1. Now, does the sequence of delta distributions $\delta_{1/n}$ converges to $\delta_1$? Almost all the people would answer no. However, does the sequence of delta distributions $\delta_{1/n}$ converges to $\delta_0$? Most people would answer yes based on the intuition that $1/n \to 0$ and so does the sequence of corresponding delta distributions, even though the support of $\delta_{1/n}$ never has any intersection with the support of $\delta_0$. Therefore, convergence can be defined for distributions not only in a point-wise way, but in a way that takes consideration of the underlying structure of the sample space.

Now returning to our adversarial divergence framework. Given an adversarial divergence $\tau$, is it possible that $\tau(\delta_1||\delta_{1/n})$ convreges to the global minimum of $\tau(\delta_1||\cdot)$? How to we define convergence to a set of points instead of only one point, in order to explain the convergence behaviour of *any* adversarial divergence? In this section we will answer these questions.

We start from two standard notions from functional analysis.

**Definition 7** (Weak-* topology on $\mathcal{P}(X)$ (see e.g. [11]))**.** *Let $X$ be a compact metric space. By associating with each $\mu \in rca(X)$ a linear function $f \longmapsto \mathbb{E}_\mu[f]$ on $C(X)$, we have that $rca(X)$*

*is the continuous dual of $C(X)$ with respect to the uniform norm on $C(X)$ (see e.g. [4]). Therefore we can equip $rca(X)$ (and therefore $\mathcal{P}(X)$) with a weak-\* topology, which is the coarsest topology on $rca(X)$ such that $\{\mu \mapsto \mathbb{E}_\mu[f] : f \in C(X)\}$ is a set of continuous linear functions on $rca(X)$.*

**Definition 8** (Weak convergence of probability measures (see e.g. [11])). *Let $X$ be a compact metric space. A sequence of probability measures $(\mu_n)$ in $\mathcal{P}(X)$ is said to weakly converge to a measure $\mu^* \in \mathcal{P}(X)$, if $\forall f \in C(X)$, $\mathbb{E}_{\mu_n}[f] \to \mathbb{E}_{\mu_*}[f]$, or equivalently, if $(\mu_n)$ is weak-\* convergent to $\mu^*$.*

The definition of weak-\* topology and weak convergence respect the topological structure of the sample space. For example, it is easy to check that the sequence of delta distributions $\delta_{1/n}$ weakly converges to $\delta_0$, but not to $\delta_1$.

Now note that Definition 8 only defines weak convergence of a sequence of probability measures to a *single* target measure. Here we generalize the definition for the single target measure to a set of target measures through *quotient topology* as follows.

**Definition 9** (Weak convergence of probability measures to a set). *Let $X$ be a compact metric space, equip $\mathcal{P}(X)$ with the weak-\* topology and let $A$ be a non-empty subspace of $\mathcal{P}(X)$. A sequence of probability measures $(\mu_n)$ in $\mathcal{P}(X)$ is said to weakly converge to the set $A$ if $([\mu_n])$ converges to $A$ in the quotient space $\mathcal{P}(X)/A$.*

With everything properly defined, we are now ready to state our convergence result. Note that an adversarial divergence is not necessarily a metric, and therefore does not necessarily induce a topology. However, convergence in an adversarial divergence can still imply some type of topological convergence. More precisely, we show a convergence result that holds for any adversarial divergence $\tau$ as long as the sample space is a compact metric space. Informally, we show that for any target probability measure, if $\tau(\mu^*||\mu_n)$ converges to the global minimum of $\tau(\mu^*||\cdot)$, then $\mu_n$ weakly converges to the set of measures that *achieve* the global minimum. Formally,

**Theorem 10.** *Let $X$ be a compact metric space, $\tau$ be an adversarial divergence over $X$, $\mu^* \in \mathcal{P}(X)$, then $\mathrm{OPT}_{\tau,\mu^*} \neq \emptyset$. Let $(\mu_n)$ be a sequence of probability measures in $\mathcal{P}(X)$. If $\tau(\mu^*||\mu_n) \to \inf_{\mu'} \tau(\mu^*||\mu')$, then $(\mu_n)$ weakly converges to the set $\mathrm{OPT}_{\tau,\mu^*}$.*

As a special case of Theorem 10, if $\tau$ is a strict adversarial divergence, i.e., $\mathrm{OPT}_{\tau,\mu^*} = \{\mu^*\}$, then converging to the minimizer of the objective function implies the usual weak convergence to the target probability measure. For example, it can be checked that the objective function of $f$-GAN is a strict adversarial divergence, therefore converging in the objective function of an $f$-GAN implies the usual weak convergence to the target probability measure.

To compare this result with our intuition, we return to the example of a sequence of delta distributions and show that as long as $\tau$ is a strict adversarial divergence, $\tau(\delta_1||\delta_{1/n})$ does not converge to the global minimum of $\tau(\delta_1||\cdot)$. Observe that if $\tau(\delta_1||\delta_{1/n})$ converges to the global minimum of $\tau(\delta_1||\cdot)$, then according to Theorem 10, $\delta_{1/n}$ will weakly converge to $\delta_1$, which leads to a contradiction.

However Theorem 10 does more than excluding undesired possibilities. It also enables us to give general statements about the structure of the class of adversarial divergences. The structural result can be easily stated under the notion of *relative strength* between adversarial divergences, which is defined as follows.

**Definition 11** (Relative strength between adversarial divergences). *Let $\tau_1$ and $\tau_2$ be two adversarial divergences, if for any sequence of probability measures $(\mu_n)$ and any target probability measure $\mu^*$, $\tau_1(\mu^*||\mu_n) \to \inf_\mu \tau_1(\mu^*||\mu)$ implies $\tau_2(\mu^*||\mu_n) \to \inf_\mu \tau_2(\mu^*||\mu)$, then we say $\tau_1$ is stronger than $\tau_2$ and $\tau_2$ is weaker than $\tau_1$. We say $\tau_1$ is equivalent to $\tau_2$ if $\tau_1$ is both stronger and weaker than $\tau_2$. We say $\tau_1$ is strictly stronger (strictly weaker) than $\tau_2$ if $\tau_1$ is stronger (weaker) than $\tau_2$ but not equivalent. We say $\tau_1$ and $\tau_2$ are not comparable if $\tau_1$ is neither stronger nor weaker than $\tau_2$.*

Not much is known about the relative strength between different adversarial divergences. If the underlying sample space is nice (e.g., subset of Euclidean space), then the variational (GAN-style) formulation of $f$-divergences using bounded continuous functions coincides with the original definition [15], and therefore $f$-divergences are adversarial divergences. [2] showed that the KL-divergence is stronger than the JS-divergence, which is equivalent to the total variation distance, which is strictly stronger than the Wasserstein-1 distance.

However, the novel fact is that we can reach the weakest strict adversarial divergence. Indeed, one implicatoin of Theorem 10 is that if $X$ is a compact metric space and $\tau$ is a strict adversarial

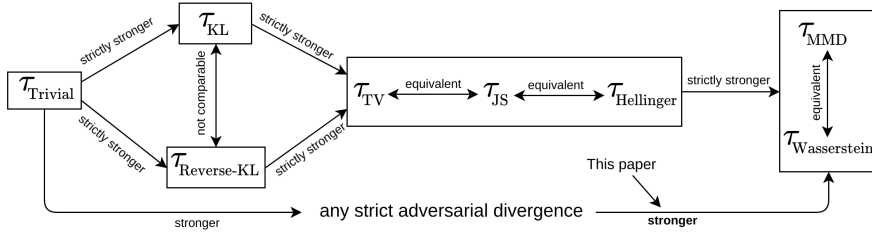

Figure 1: Structure of the class of strict adversarial divergences

divergence over $\tau$, then $\tau$-convergence implies the usual weak convergence on probability measures. In particular, since the Wasserstein distance metrizes weak convergence of probability distributions (see e.g. [14]), as a direct consequence of Theorem 10, the Wasserstein distance is in the equivalence class of the weakest strict adversarial divergences. In the other direction, there exists a trivial strict adversarial divergence

$$\tau_{\text{Trivial}}(\mu||\nu) \triangleq \begin{cases} 0, & \text{if } \mu = \nu, \\ +\infty, & \text{otherwise,} \end{cases} \tag{7}$$

that is stronger than any other strict adversarial divergence. We now incorporate our convergence results with some previous results and get the following structural result.

**Corollary 12.** *The class of strict adversarial divergences over a bounded and closed subset of a Euclidean space has the structure as shown in Figure 1, where $\tau_{Trivial}$ is defined as in (7), $\tau_{MMD}$ is corresponding to example (c) in Section 3, $\tau_{Wasserstein}$ is corresponding to example (d) in Section 3, and $\tau_{KL}$, $\tau_{Reverse\text{-}KL}$, $\tau_{TV}$, $\tau_{JS}$, $\tau_{Hellinger}$ are corresponding to example (b) in Section 3 with $f(x)$ being $x\log(x)$, $-\log(x)$, $\frac{1}{2}|x-1|$, $-(x+1)\log(\frac{x+1}{2}) + x\log(x)$, $(\sqrt{x}-1)^2$, respectively. Each rectangle in Figure 1 represents an equivalence class, inside of which are some examples. In particular, $\tau_{Trivial}$ is in the equivalence class of the strongest strict adversarial divergences, while $\tau_{MMD}$ and $\tau_{Wasserstein}$ are in the equivalence class of the weakest strict adversarial divergences.*

# 6    Related Work

There has been an explosion of work on GANs over the past couple of years; however, most of the work has been empirical in nature. A body of literature has looked at designing variants of GANs which use different objective functions. Examples include [10], which propose using the f-divergence between the target $\mu$ and the generated distribution $\nu$, and [5, 9], which propose the MMD distance. Inspired by previous work, we identify a family of GAN-style objective functions in full generality and show general properties of the objective functions in this family.

There has also been some work on comparing different GAN-style objective functions in terms of their convergence properties, either in a GAN-related setting [2], or in a general IPM setting [12]. Unlike these results, which look at the relationship between several specific strict adversarial divergences, our results apply to an entire class of GAN-style objective functions and establish their convergence properties. For example, [2] shows that KL-divergnce, JS-divergence, total-variation distance are all stronger than the Wasserstein distance, while our results generalize this part of their result and says that any strict adversarial divergence is stronger than the Wasserstein distance and its equivalences. Furthermore, our results also apply to non-strict adversarial divergences.

That being said, it does not mean our results are a complete generalization of the previous convergence results such as [2, 12]. Our results do not provide any methods to compare two strict adversarial divergences if none of them is equivalent to the Wasserstein distance or the trivial divergence. In contrast, [2] show that the KL-divergence is stronger than the JS-divergence, which is equivalent to the total variation distance, which is strictly stronger than the Wasserstein-1 distance.

Finally, there has been some additional theoretical literature on understanding GANs, which consider orthogonal aspects of the problem. [3] address the question of whether we can achieve generalization bounds when training GANs. [13] focus on optimizing the estimating power of kernel distances. [5] study generalization bounds for MMD-GAN in terms of fat-shattering dimension.

## 7  Discussion and Conclusions

In conclusion, our results provide insights on the cost or loss functions that should be used in GANs. The choice of cost function plays a very important role in this case – more so, for example, than data domains or network architectures. For example, most works still use the DCGAN architecture, while changing the cost functions to achieve different levels of performance, and which cost function is better is still a matter of debate. In particular we provide a framework for studying many different GAN criteria in a way that makes them more directly comparable, and under this framework, we study both approximation and convergence properties of various loss functions.

## 8  Acknowledgments

We thank Iliya Tolstikhin, Sylvain Gelly, and Robert Williamson for helpful discussions. The work of KC and SL were partially supported by NSF under IIS 1617157.

## Footnotes

[1]Weakness is actually a desirable property since it prevents the divergence from being too discriminative (saturate), thus providing more information about how to modify the model to approximate the true distribution.

[2] To the best of our knowledge, neither (3) or (4) was used in any GAN algorithm. However, since our focus in this paper is not implementing new algorithms, we leave experiments with this formulation for future work.

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
