[Supplementary Material]

# A  Proof of Theorem 4

We observe that the assumptions of the theorem imply that for any $\mu \in \mathcal{P}(X)$,

$$\tau(\mu||\mu) = \sup_{\theta \in \Theta} \mathbb{E}_{\mu \otimes \mu} [-r_\theta] = \mathbb{E}_{\mu \otimes \mu} \left[ -r_{\theta_\mu^\mu} \right] = -c. \tag{8}$$

The assumptions also imply that for any $\mu, \nu \in \mathcal{P}(X)$,

$$\begin{aligned}
\tau(\mu||\nu) &\geq \mathbb{E}_{\mu \otimes \nu} \left[ m_{\theta_\nu^\mu} - r_{\theta_\nu^\mu} \right] \\
&\geq \mathbb{E}_{\mu \otimes \nu} \left[ -r_{\theta_\nu^\mu} \right] \\
&= -c
\end{aligned}$$

Fix $\mu^*, \mu \in \mathcal{P}(X)$ and assume $\mu \in \mathcal{M}_{\mu^*}$, i.e. $\mathbb{E}_{\mu^*}[v_\theta] = \mathbb{E}_\mu[v_\theta]$ for any $\theta \in \Theta$. Then

$$\begin{aligned}
\tau(\mu^*||\mu) &= \sup_{\theta \in \Theta} \mathbb{E}_{\mu^* \otimes \mu} [m_\theta - r_\theta] \\
&= \sup_{\theta \in \Theta} \mathbb{E}_{\mu^* \otimes \mu} [-r_\theta] \\
&= \mathbb{E}_{\mu^* \otimes \mu} \left[ -r_{\theta_\mu^{\mu^*}} \right] \tag{9} \\
&= -c. \tag{10}
\end{aligned}$$

Therefore $\mu \in \mathrm{OPT}_{\tau,\mu^*}$.

# B  Proof of Theorem 5

Since by Theorem 4 we already have $\mathcal{M}_{\mu^*} \subset \mathrm{OPT}_{\tau,\mu^*}$, we only need to prove for any $\mu^* \in \mathcal{P}(X)$,

$$\mathcal{P}(X) \setminus \mathcal{M}_{\mu^*} \subseteq \mathcal{P}(X) \setminus \mathrm{OPT}_{\tau,\mu^*}.$$

Fix $\mu^*, \mu \in \mathcal{P}(X)$. Assume there exists $\theta' \in \Theta$ such that $\mathbb{E}_{\mu^* \otimes \mu}[m_{\theta'}] \neq 0$. If $\mathbb{E}_{\mu^* \otimes \mu} \left[ m_{\theta_\mu^{\mu^*}} \right] \neq 0$, then we have $\mathbb{E}_{\mu^* \otimes \mu} \left[ m_{\theta_\mu^{\mu^*}} \right] > 0$. Now note that

$$\begin{aligned}
\tau(\mu^*||\mu) &= \sup_{\theta \in \Theta} \mathbb{E}_{\mu^* \otimes \mu} [m_\theta - r_\theta] \\
&\geq \mathbb{E}_{\mu^* \otimes \mu} \left[ m_{\theta_\mu^{\mu^*}} - r_{\theta_\mu^{\mu^*}} \right] \\
&= \mathbb{E}_{\mu^* \otimes \mu} \left[ m_{\theta_\mu^{\mu^*}} \right] - \mathbb{E}_{\mu^* \otimes \mu} \left[ r_{\theta_\mu^{\mu^*}} \right] \\
&> -\mathbb{E}_{\mu^* \otimes \mu} \left[ r_{\theta_\mu^{\mu^*}} \right] \\
&= -c \\
&= \tau(\mu^*||\mu^*),
\end{aligned}$$

where the last equality is due to (8). Thus $\mu \notin \mathrm{OPT}_{\tau,\mu^*}$. For the rest of the proof we assume $\mathbb{E}_{\mu^* \otimes \mu} \left[ m_{\theta_\mu^{\mu^*}} \right] = 0$. Then by (6) we have $\nabla_{\theta_\mu^{\mu^*}} \mathbb{E}_{\mu^* \otimes \mu} [m] \neq \mathbf{0}$. Also because $\theta_\mu^{\mu^*}$ is an interior point of $\Theta$ and $\theta_\mu^{\mu^*}$ is a minimizer of $\theta \mapsto \mathbb{E}_{\mu^* \otimes \mu}[r_\theta]$, by Fermat's stationary points theorem, we have $\nabla_{\theta_\mu^{\mu^*}} \mathbb{E}_{\mu^* \otimes \mu} [r] = \mathbf{0}$. Therefore $\nabla_{\theta_\mu^{\mu^*}} \mathbb{E}_{\mu^* \otimes \mu} [m - r] \neq \mathbf{0}$. Thus again by Fermat's stationary points theorem there exists a $\theta' \in \Theta$ such that

$$\begin{aligned}
\mathbb{E}_{\mu^* \otimes \mu} [m_{\theta'} - r_{\theta'}] &> \mathbb{E}_{\mu^* \otimes \mu} \left[ m_{\theta_\mu^{\mu^*}} - r_{\theta_\mu^{\mu^*}} \right] \\
&\geq \mathbb{E}_{\mu^* \otimes \mu} \left[ -r_{\theta_\mu^{\mu^*}} \right] \\
&= -c \\
&= \tau(\mu^*||\mu^*),
\end{aligned}$$

where the last equality is due to (8). Finally note that

$$\tau(\mu^*||\mu) \geq \mathbb{E}_{\mu^* \otimes \mu} [m_{\theta'} - r_{\theta'}] > \tau(\mu^*||\mu^*).$$

Therefore $\mu \notin \mathrm{OPT}_{\tau,\mu^*}$. This concludes the proof.

## C   Proof of Corollary 6

Recall that by assumption $f^*(x) \geq x$ for any $x \in \mathbb{R}$ and $f^*(x_0) = x_0$ for some $x_0 \in \text{int}\,(\text{dom}\, f^*)$. Since $f^*$ is continuously differentiable on $\text{int}\,(\text{dom}\, f^*)$ necessarily we have $(f^*)'(x_0) = 1$.

For each $\theta \in \Theta$, let $v_\theta(x) = \theta^\mathsf{T}(\psi(x), 1)$, $m_\theta(x, y) = v_\theta(x) - v_\theta(y)$, $r_\theta(x, y) = f^*(\theta^\mathsf{T}(\psi(y), 1)) - \theta^\mathsf{T}(\psi(y), 1) \geq 0$, then $\tau(\mu \| \nu) = \sup_{\theta \in \Theta} \mathbb{E}_{\mu \otimes \nu}[m_\theta - r_\theta]$. $v_\theta$ and $r_\theta$ are bounded continuous functions since both $f^*$ and $\psi$ are continuous functions, $\theta^\mathsf{T}(\psi(x), 1) \in \text{dom}\, f^*$ for any $(x, \theta) \in X \times \Theta$, and $X$ is a compact set. Let $\theta_\nu^\mu = (0, 0, \cdots, 0, x_0)$, that is, a vector whose last coordinate is $x_0$ and 0 elsewhere. We have that for any $\mu^*, \mu \in \mathcal{P}(X)$

$$\tau(\mu^* \| \mu) \geq \mathbb{E}_{\mu^* \otimes \mu}\left[m_{\theta_\mu^{\mu^*}} - r_{\theta_\mu^{\mu^*}}\right] = \mathbb{E}_{\mu^* \otimes \mu}[f(x_0) - x_0] = 0$$

$$\tau(\mu^* \| \mu^*) = \sup_{\theta \in \Theta} \mathbb{E}_{\mu^* \otimes \mu}[-r_\theta] \leq 0$$

Thus $\text{OPT}_{\tau, \mu^*} = \{\mu : \tau(\mu^* \| \mu) = 0\} \ni \mu^*$. It remains to show $\text{OPT}_{\tau, \mu^*} = \{\mu : \mathbb{E}_{\mu^*}[\psi] - \mathbb{E}_\mu[\psi]\}$.

Because $x_0$ is an interior point of $\text{dom}\, f^*$, we have $\theta_\nu^\mu$ is an interior point of $\Theta$, due to the compactness of $X$ and all $\psi_i$ being continuous and therefore bounded continuous. Also, it is easy to see that $r_{\theta_\nu^\mu}$ is a constant function.

Because $f^*(x) \geq x$ for any $x \in \mathbb{R}$, we have that $r_\theta(x, y) \geq 0$ for any $\theta \in \Theta$ and $x, y \in X$. On the other hand, we have $r_{\theta_\nu^\mu}(x, y) = 0$ for any $x, y \in X$. Therefore $r_{\theta_\nu^\mu} \leq r_\theta$ for any $\theta \in \Theta$. Also, it is easy to see that $\mathbb{E}_{\mu \otimes \nu}[m_{\theta_\nu^\mu}] = 0$.

Now it suffices to show that $\theta \mapsto \mathbb{E}_{\mu \otimes \nu}[m_\theta]$ and $\theta \mapsto \mathbb{E}_{\mu \otimes \nu}[r_\theta]$ both has gradient at $\theta_\nu^\mu$ and condition (6) in Theorem 5 hold.

Because both $\psi$ and $f^*$ are continuously differentiable and $X$ is a compact space, according to the Leibniz rule for differentiating an integral in general measurable spaces (see e.g., [1] ), $\theta \mapsto \mathbb{E}_{\mu \otimes \nu}[r_\theta]$ has gradient at $\theta_\nu^\mu$. Also note

$$\nabla_{\theta_\nu^\mu} \mathbb{E}_{\mu \otimes \nu}[m] = \mathbb{E}_{x \sim \mu,\, y \sim \nu}[(\psi(x) - \psi(y), 0)] = (\mathbb{E}_\mu[\psi] - \mathbb{E}_\nu[\psi], 0) \tag{11}$$

To verify condition (6), note that if (11) is equal to $\mathbf{0}$, then $\mathbb{E}_\mu[\psi] = \mathbb{E}_\nu[\psi]$, therefore for any $\theta \in \Theta$, $\mathbb{E}_{\mu \otimes \nu}[m_\theta] = \theta^\mathsf{T}(\mathbb{E}_\mu[\psi] - \mathbb{E}_\nu[\psi], 0) = 0$. The proof is concluded.

## D   Proof of Theorem 10

We first need a standard result in functional analysis. A brief proof is provided for completeness.

**Lemma 13.** *If $X$ is a compact metric space, then $\mathcal{P}(X)$ is weak-\* compact.*

*Proof.* By the Banach-Alaoglu theorem, the following closed unit ball is weak-\* compact.

$$\{\mu \in rca(X) : |\mu|(X) \leq 1\}. \tag{12}$$

Since the constant function $\mathbf{1}$ is in $C(X)$. The following set is weak-\* closed.

$$\{\mu \in rca(X) : \mu(X) = 1\} = \{\mu \in rca(X) : \langle \mathbf{1}, \mu \rangle = 1\}. \tag{13}$$

We also claim that

$$\{\mu \in rca(X) : \mu(A) \geq 0 \text{ for every Borel set in } X\} = \bigcap_{f \in C_+(X)} \{\mu \in rca(X) : \langle f, \mu \rangle \geq 0\}, \tag{14}$$

which is weak-\* closed. To justify the claim, on one hand, the l.h.s. is clearly a subset of the r.h.s.; on the other hand, to show the r.h.s. is also a subset of the l.h.s., consider a $\mu \in rca(X)$ with a Borel set $A$ such that $\mu(A) < 0$ (i.e., $\mu$ is not in the l.h.s.), then by Lusin's Theorem the measurable function $\mathbf{1}_A$ can be approximated by functions in $C(X)$ in the sense that for any $\epsilon > 0$, there exists a $f_\epsilon \in C(X)$ such that

$$|\mathbb{E}_\mu[\mathbf{1}_A] - \mathbb{E}_\mu[f_\epsilon]| < \epsilon,$$

Choose $\epsilon = \frac{|\mu(A)|}{2}$ we get a function $f_\epsilon' = \max(f_\epsilon, \mathbf{0})$ in $C(X)$ such that $\mathbb{E}_\mu[f_\epsilon'] \leq 0$. This means $\mu$ is not in the r.h.s. Note that

$$\mathcal{P}(X) = (12) \cap (13) \cap (14).$$

Since the intersection of a compact subset and a closed subset is a compact subset, we conclude that $\mathcal{P}(X)$ is weak-\* compact. $\qquad\square$

Now we can start the main proof. We equip $\mathcal{P}(X)$ with the weak-* topology. Let $\mu^* \in \mathcal{P}(X)$. Note that the function $\tau(\mu^*||\cdot)$ is the supremum of a family of affine continuous functions on $\mathcal{P}(X)$, therefore $\tau(\mu^*||\cdot)$ is lower semi-continuous on $\mathcal{P}(X)$. Note that by Lemma 13, $\mathcal{P}(X)$ is compact. Therefore by Weierstrass extreme value theorem, $\tau(\mu^*||\cdot)$ attains its minimual value on $\mathcal{P}(X)$, therefore $\text{OPT}_{\tau,\mu^*} \neq \emptyset$.

Let $(\mu_n)$ be a sequence in $\mathcal{P}(X)$. Assume $\tau(\mu^*||\mu_n) \to \inf_{\mu'} \tau(\mu^*||\mu')$, we need to show that in the quotient space $\mathcal{Q} = \mathcal{P}(X)/\text{OPT}_{\tau,\mu^*}$, $([\mu_n])$ converges to $\text{OPT}_{\tau,\mu^*}$. Let $\mathcal{N}$ be any open neighbourhood of $\text{OPT}_{\tau,\mu^*}$ in $\mathcal{Q}$. We need to show that $([\mu_n])$ is eventually in $\mathcal{N}$.

First we show that $\mathcal{Q} \setminus \mathcal{N}$ is compact. By Lemma 13, $\mathcal{P}(X)$ is compact. Since $\mathcal{Q}$ is a quotient space of $\mathcal{P}(X)$, $\mathcal{Q}$ is compact. Observe that $\mathcal{Q} \setminus \mathcal{N}$ is a closed subset of $\mathcal{Q}$, therefore $\mathcal{Q} \setminus \mathcal{N}$ is compact.

Recall that $\tau(\mu^*||\cdot)$ is lower semi-continuous on $\mathcal{P}(X)$. Now observe that $\tau(\mu^*||\cdot)$ is also a function on $\mathcal{Q}$, and since $\mathcal{Q}$ is a quotient space of $\mathcal{P}(X)$, $\tau(\mu^*||\cdot)$ is also lower semi-continuous on $\mathcal{Q}$. By Weierstrass extreme value theorem, there exists $[\mu'] \in \mathcal{Q} \setminus \mathcal{N}$ such that

$$\tau(\mu^*||[\mu']) = \inf_{[\mu] \in \mathcal{Q} \setminus \mathcal{N}} \tau(\mu^*||[\mu])$$

Since $\text{OPT}_{\tau,\mu^*} \notin \mathcal{Q} \setminus \mathcal{N}$, we have $[\mu'] \neq [\text{OPT}_{\tau,\mu^*}]$. Therefore $\tau(\mu^*||[\mu']) > \inf_{\mu} \tau(\mu^*||\mu)$. Recall that $\tau(\mu^*||[\mu_n]) \to \inf_{\mu} \tau(\mu^*||\mu)$, $\tau(\mu^*||[\mu_n])$ will be eventually less than $\tau(\mu^*||[\mu'])$. This means $([\mu_n])$ will eventually be in $\mathcal{N}$.

# E   Proof of Corollary 12

It is known that for nice spaces (e.g., bounded and closed subset of a Euclidean space), the variational (GAN-style) formulation of $f$-divergences using bounded continuous functions is equivalent to the original definition [15]. Therefore we them interchangeably. [2] already showed that total variation is equivalent to JS divergence and they are not equivalent to the Wasserstein distance. These two are also known to be equivalent to the squared Hellinger distance by noticing that

$$\tau_{\text{Hellinger}}(\mu||\nu) \leq \tau_{\text{TV}}(\mu||\nu) \leq \sqrt{2\tau_{\text{Hellinger}}(\mu||\nu)}.$$

Both KL and Reverse-KL divergence are stronger than total variation by Pinsker's inequality. They are in fact strictly stronger than total variation. Let $\mu^* = U(0,1)$, $\mu_n = U(1/n, 1 + 1/n)$, where $U(a,b)$ is the uniform distribution on $(a,b)$. Note $\tau_{\text{KL}}(\mu^*||\mu_n) = \tau_{\text{Reverse-KL}}(\mu^*||\mu_n) = +\infty$ for any $n$ while $\tau_{\text{TV}}(\mu^*||\mu_n) \to 0$. We can also show they are not comparable with each other by considering $\mu_n = U(0, 1 - 1/n)$ and $\mu_n = U(0, 1 + 1/n)$ while $\mu^*$ is still $U(0,1)$. The same examples also show they are strictly weaker than the trivial divergence.

It is also known that $\tau_{\text{Wasserstein}}$ and $\tau_{\text{MMD}}$ metrize the weak-* topology of $\mathcal{P}(X)$ if $X$ is a compact metric space (see, e.g., [12]), therefore by Theorem 10 they are in the equivalence class of the weakest strict adversarial divergences.

It remains to show the trivial divergence is stronger than any strict adversarial divergence. Any sequence $\mu_n$ converging to $\mu^*$ under the trivial divergence is eventually $\mu^*$, therefore trivially converges under any other strict adversarial divergence.