[Reviews · NeurIPS 2017]

Reviewer 1



The authors present a formal analysis to characterize general adversarial learning. The analysis shows that under certain conditions on the objective function the adversarial process has a moment-matching effect. They also show results on convergence properties. The writing is quite dense and may not be accessible to most of the NIPS audience. I did not follow the full details myself. I defer to reviewers closer to the area. One general reservation I have about a formal analysis along the line presented is whether the results actually provide any insights useful in the practice of generative adversarial networks. In particular, do the results suggest properties of domains / distributions that make such domains suitable for a GAN approach, and, conversely, what practical domain properties would suggest a bad fit for GAN? Do the results provide guidelines for the choice of deep net architecture or learning parameters? Ideally, a formal analysis provides such type of feedback to the practitioners. If no such feedback can be provided, the formal results may still be of interest but unfortunately, they will remain largely a theoretical exercise. But at least, I would like to see a brief discussion on the impact on the practice of these results because it's the practical success of the GAN approach that has made it so exciting. (A similar issue arises with any formal framework that "explains" the success of deep learning. Such explanations become much more interesting if they also identify domain properties that predict where deep learning would work well and where it would not, in relation to real-world domains. I.e. what are the lessons for the practice of deep learning?) I'm happy with the authors' response. It would be great to see the feedback incorporated in the revised paper. I've raised my rating.

Reviewer 2



The paper studies of the convergence properties of a large family of Generative Adversarial Networks. The authors introduce the concept of adversarial divergence \tau(\mu||\nu), which is defined as the supremum of the expectation of some convex function f under the product measure of \mu and \nu. It is shown in the paper that the objective functions of a large family of GAN variations in the literature can be formulated as adversarial divergences. The adversarial divergence is characterised as strict if there is a unique minimiser \nu=\mu for \tau(\mu||\nu), given the target distribution \mu. If case of a non-strict divergence (i.e. more than one minimising distribution exist) it is shown that minimising \tau(\mu||\nu) has a moment matching effect for \mu and \nu, under some additional assumptions. Finally, the authors show that converge in \tau(\mu||\nu) implies also convergence towards the set of optimising distributions for \tau(\mu||\nu), given that the sample space is a compact metric space. I think that the paper makes some interesting contributions to the theory of GANs. The formulation of a large family of objective functions in the literature as adversarial divergences is well-demonstrated, and the convergence results provide useful insights on the properties of the different GAN approaches.

Reviewer 3



This paper mainly addresses two important questions in generative adversarial learning by the introduced adversarial divergence: the first is to relate the optimality of divergence to certain matching conditions under generalized moment matching framework; the second is to discuss the weak convergence and relative strengthen of different divergences. In general, the presentation in this paper provides a relatively deep understanding of GAN-like models. It will benefit the research in this community, and point a theoretical way for future working, rather than most of the heuristic tricks, so I vote for accept. Some other comments: 1. Line 111: (f) can you provide an example or short description of transportation cost function in the appendix for readers better understanding? 2. Line 149: Does the assumption of theorem 4 hold for (f)? 3. Line 210: typo, converges 4. Can the property of convergence provide some practical guide in GAN training? Like which distance is a better choice for loss function? 5. Although this is a theoretical paper, I hope the authors can support their claims with some experimental results